# Effect of Ammonium Sulfide on Sulfidization Flotation of Malachite

Ayman M. Ibrahim [1,2], Xiaodong Jia [1], Chao Su [1], Jinpeng Cai [1], Peilun Shen [1,*] and Dianwen Liu [1,*]

[1] State Key Laboratory of Complex Nonferrous Metal Resources Clean Utilization, Yunnan Key Laboratory of Green Separation and Enrichment of Strategic Mineral Resources, Faculty of Land Resource Engineering, Kunming University of Science and Technology, Kunming 650093, China
[2] Department of Mining Engineering, Faculty of Engineering Sciences, University of Nyala, Nyala 63311, Sudan
* Correspondence: peilunshen@kust.edu.cn (P.S.); dianwenliu@kust.edu.cn (D.L.)

**Abstract:** Recently, several studies have shown the positive effect of sulfidization flotation on malachite surfaces and its enhancing methods. Therefore, this paper was focused on the effect of ammonium sulfide and sodium sulfide on the sulfidization of malachite, respectively; this was investigated using different devices such as the micro-flotation tests, Zeta potential measurements, ToF–SIMS, XPS analysis, and FTIR. Thus, Fourier transform infrared spectroscopy results demonstrated that a new characteristic peak of Cu-S bonds was formed and adsorbed on malachite surfaces at 1694 cm$^{-1}$, as confirmed by XPS analysis. Notably, malachite with ammonium sulfide ions had a significantly higher flotation recovery than malachite with an excess of sodium sulfide ions, as concerns of sulfidization types. Conclusively, all the experiments in this study confirmed that additional copper sulfide products were formed on the malachite surface, increasing the hydrophobicity of the malachite.

**Keywords:** malachite; sulfidization; ammonium sulfide; flotation factors





## 1. Introduction

The development, or exploitation, of copper oxide ores is making it increasingly difficult to maintain sufficient levels of copper sulfides to keep up with the rising demand for copper resources [1,2]. Most valuable copper sulfide minerals are related to copper oxide minerals [3]. Malachite is a copper oxide mineral found in primary copper ore deposits. Froth flotation is being widely used to pre-concentrate copper minerals prior to leaching, to enable more environmentally friendly metallurgical extractions [4,5].

The attachment of particles to bubble surfaces is an essential interaction step during the flotation process since the interaction with particles and bubbles provides the highest efficiency of the process [6,7]. Even though the direct flotation technique is valuable, its commercial viability is restrained due to the reduced selectivity of gangue minerals. In malachite flotation, the most common approach is to use xanthate collectors in the existence of sulfidizing reagents for floating the malachite minerals. In the sulfidation flotation method, $Na_2S$, $(NH_4)_2SO_4$, and $(NH_4)_2S$ are often used to activate the malachite as sulfidizing reagents and facilitate their recovery [8,9]. The most common technique to overcome that problem is to sulfidize the surface of oxide-copper minerals before the collector addition [10].

Generally, inside the flotation of malachite, xanthates are used as collectors and are mixed with $Na_2S$ or $(NH_4)_2S$ as the sulfidation reagents. However, $Na_2S$ or $(NH_4)_2S$ are risky with the presence of oxygen and may release $H_2S$ into the air, which is potentially harmful to the environment as well as to humans [9,11]. Non-sulfide copper minerals' reaction to sulfidizing agents such as $NaHS$, $NH_4HS$, or $Na_2S$ was an undeniable defect in the sulfidation flotation method [12,13].

In this study, the goal was to investigate the effect of the sulfurization process using different types of sulfidizer reagents, such as ammonium sulfide and sodium sulfide, on the flotation recovery of malachite, and also to investigate the cumulative role of sulfide

ion adsorption on malachite flotation in order to understand the underlying mechanism. Moreover, the (NH$_4$)$_2$S was examined as both a sulfidizer and an activator in order to interpret the feasibility of sulfur adsorption on the malachite surface. The effectiveness of reagents to comply with high reagent consumption and environmental aspects was studied. The variability in the effect of sulfur adsorption between sodium sulfide and ammonium sulfide was compared by normalization peak intensity. In addition, to evaluate the availability of the results, micro-flotation experiments, FTIR analysis, Zeta, and XRD analysis were performed within this investigation.

## 2. Methods

### 2.1. Materials and Reagents

Samples of pure malachite (Cu$_2$CO$_3$(OH)$_2$) were acquired from Yunnan, China, then crushed in a double roll laboratory crusher before being ground in an agate mortar separately. Then, dry-sieved samples were used to collect size fractions between $-75 + 38$ μm of malachite for micro-flotation experiments, IR spectroscopic, XPS and ToF–SIMS analyses. The analysis results of XRF (Table 1) revealed that the sample contained 56.14% Cu, and no impurity peak was found except for the diffraction peaks of malachite through the X-ray diffraction results of malachite samples in Figure 1. The results showed that the purity of malachite met the requirements of micro-flotation experiments. For zeta potential measurements, $-5$ μm fractions were selected.

**Table 1.** The analysis of malachite composition by XRF.

| Elements | Cu | Mn | Fe | SiO$_2$ | Al$_2$O$_3$ | CaO | MgO | Others |
|---|---|---|---|---|---|---|---|---|
| **Wt.%** | 56.14 | 0.400 | 0.100 | 2.320 | 0.560 | 0.700 | 0.321 | 39.459 |

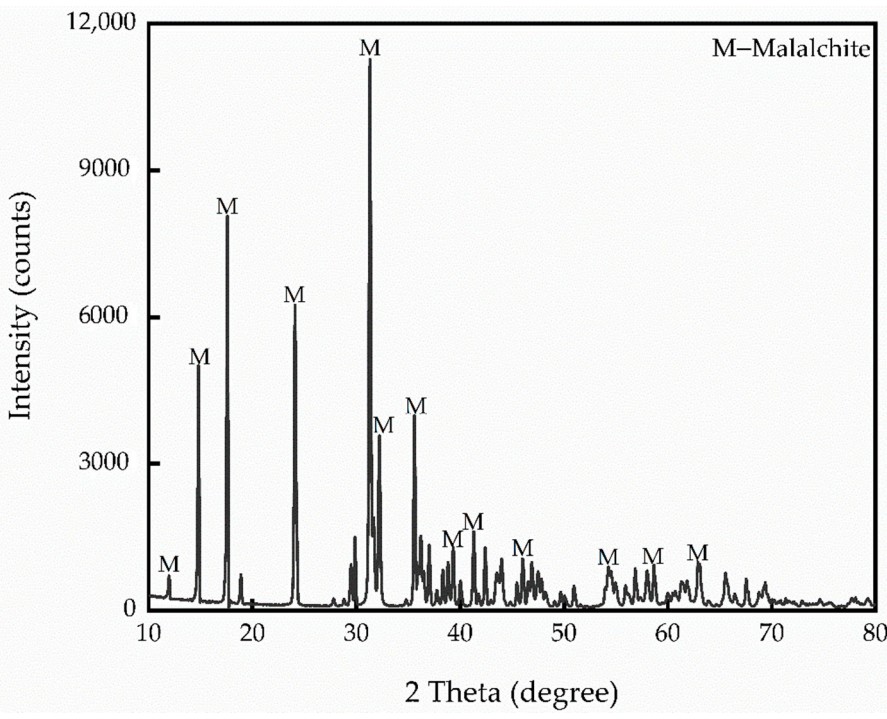

**Figure 1.** X-ray diffraction of malachite sample.

For pH modification, an analytical grade of sodium hydroxide (NaOH) and hydrochloric acid was chosen. (NH$_4$)$_2$S and Na$_2$S·9H$_2$O have been used as sulfidizing agents, and the concentration of ammonium sulfide used was 14% in water. The commercial grade of sodium-butyl xanthate (NaBX) was used as a collector. All of the experiments were carried out with distilled water.

## 2.2. Micro-Flotation Experiments

Micro-flotation experiments were performed in a modified Hallimond tube with a capacity of 50 mL. In each flotation test, 3 g of the fine malachite sample became conditioned for 3 min in a regarded quantity of deionized water in a 50 mL beaker, and the impeller velocity was held at a steady l750 rpm. After that, pH regulators (stirring for 1 min) and the pH of the pulp were adjusted to around 8.7. Flotation reagents were injected into the pulp in turn; an $(NH_4)_2S$ and $Na_2S$ solution of $6 \times 10^{-2}$ mol/L and $6 \times 10^{-2}$ mol/L, respectively, was added to the sulfuring of the malachite surface for 3 min as well as NaBX and MIBC (if needed, stirring for 3 min). The conditioned slurry was added to the Hallimond tube, where it floated for 10 min at a rate of around 10 mL/min of nitrogen gas. Then, float and sink were filtered and dried, and the weight distribution between the two solid items was calculated using Equation (1). All flotation experiments were conducted at 25 °C room temperature:

$$\varepsilon = \frac{m_1}{m_1 + m_2} \times 100\% \tag{1}$$

where $m_1$ and $m_2$ are the foam and tailing product recoveries, respectively.

## 2.3. Zeta Potential Measurements

Zeta potential measurements were carried out using the ZetaPlus analyzer (Brookhaven Instruments, Holtsville, NY, USA). In each test, 0.5 g of natural mineral samples with diameters of −5 μm was added to 40 mL $1.0 \times 10^{-3}$ mol/L $KNO_3$ solution, and then the pulp's pH was regulated to the anticipated value with diluted HCl or NaOH solutions. Freshly equipped $Na_2S \cdot 9H_2O$ solutions of $6 \times 10^{-2}$ mol/L and $(NH_4)_2S$ $6 \times 10^{-2}$ mol/L were added to the mixture (if needed, stirring for 3 min); Zeta potential measurements of malachite particles were independently measured six times in the presence or absence of $Na_2S$, and the average of zeta potential and the usual deviation were calculated.

## 2.4. FTIR Spectroscopy Analysis

The experiments were conducted with FTIR from Nicolet, USA. Fresh $Na_2S$ and $(NH_4)_2S$ solutions of $6 \times 10^{-2}$ mol/L were prepared, respectively. FTIR spectra were performed in the following manner: 0.5 g of the malachite sample with a particle size less than 5 μm was added to a 100 mL volumetric flask, and 50 mL of deionized water was added. After that, the pH value was modified to 8.7, and then $Na_2S$ and $(NH_4)_2S$ solutions $6 \times 10^{-2}$ mol/L were added as needed. After stirring for 10 min, malachite samples were filtered and dried in a vacuum dryer at 60 °C. Ultimately, FTIR experiments were conducted on the malachite samples.

## 2.5. XPS Analysis

The XPS analysis was conducted employing a PHI5000 Versa Probe II (PHI5000, ULVAC-PHI Inc., Chigasaki, Kanagawa, Japan). To determine all present components, the analyzed specimens were exposed to survey examination. The recorded spectra were calibrated using the C1s peak at 284.8 eV as a reference. The data were compiled and treated from the peak area using MultiPak software. Samples were prepared for XPS measurements as follows: 5 g of malachite was added to a 50 mL aqueous solution, and the slurry was incubated at 298 K with the mineral mixture as needed. Freshly prepared ammonium sulfide, $6 \times 10^{-2}$ mol/L, or sodium sulfide, $6 \times 10^{-2}$ mol/L, were added with MIBX solutions of $1 \times 10^{-5}$ mol/L to sulfidize the mineral's surface for 3 min at a natural pH before XPS analysis. Malachite samples were filtered, rinsed three times with distilled water, and dried under a vacuum at room temperature within the silica gel desiccator.

## 2.6. ToF–SIMS Analysis

Malachite surface composition was characterized using ToF–SIMS IV (ION-TOF, Munster, Germany). Malachite samples were ultrasonically washed and then transferred to a beaker. Freshly prepared $(NH_4)_2S$ and $Na_2S$ were added to the solution based on the

regents needed for the ToF–SIMS analysis and subsequently placed directly in 50 mL of $6 \times 10^{-2}$ mol/L ammonium sulfide solution and submerged for 2 h at 25 °C.

The powder of the malachite sample was attached with a double conducting adhesive, and nitrogen gas was used to detach the fine particles from the surface of the granular particles. After that, the samples were moved to the device sample loading room, and thus, the selected analysis region was 50 μm × 50 μm, with 3 pulse widths for every point, and were tested in a negative ion mode at 500 μm × 500 μm. The malachite samples were then washed with distilled water, dried with highly pure $N_2$, and analyzed with ToF–SIMS.

## 3. Results and Discussions

### 3.1. Flotation Findings

Figure 2 shows the flotation recovery of the malachite when sulfidized with $Na_2S$ and $(NH_4)_2S$ as a variable of NaBX concentrations. With different dosages of NaBX concentrations, respectively, a low flotation recovery was acquired when only $Na_2S$ was used, while malachite sulfidized with $(NH_4)_2S$ had a considerably greater average recovery rate than the malachite sulfidized with $Na_2S$, suggesting that due to variations in the quantity of xanthates adsorbed on the surface in this case, the malachite was treated with ammonium sulfide [14].

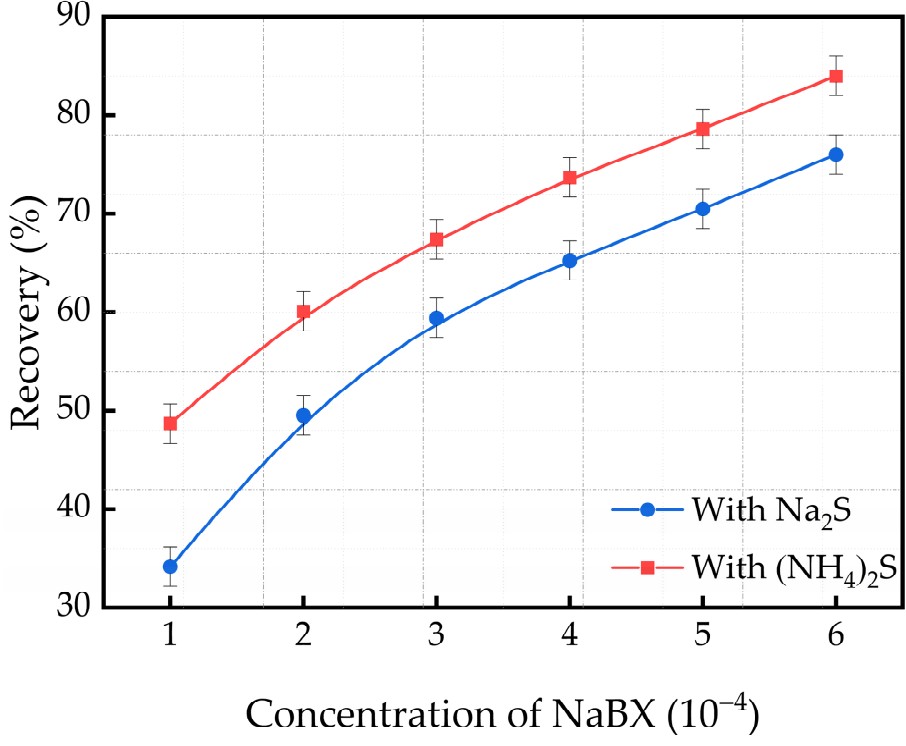

**Figure 2.** Flotation recovery of sulfidized malachite with $(NH_4)_2S$ and $Na_2S$ as a function of NaBX concentrations: $Na_2S = 6 \times 10^{-2}$ mol/L; $(NH_4)_2S = 6 \times 10^{-2}$ mol/L; NaBX = $2 \times 10^{-4}$ M; pH = $8.7 \pm 0.05$.

### 3.2. Zeta Potential Results

Figure 3 shows the zeta potential of (a) malachite, (b) malachite + $Na_2S$ and (c) malachite + $(NH_4)_2S$ as a variable of pH. The isoelectric malachite point is about 8.6. It can be found that the malachite surface curve moves in a negative direction when the $Na_2S$ is added, and the IEP is decreased to a pH < 8.3. This is because the $HS^-$ and $S^{2-}$ ions are produced by hydrolyzed $Na_2S$ attached to metal ions on a malachite surface. After adding ammonium sulfide, the malachite surface potential moved in a negative direction, and the IEP shifted to a pH of 8.1, indicating that more sulfur species were transferred from the solution to the malachite surface, implying the formation of increasing sulfurized products. After that, it might be adsorbed on a negative mineral surface through electrostatic adsorp-

tion. This result is consistent with the findings of malachite micro-flotation and dissolution experiments [15].

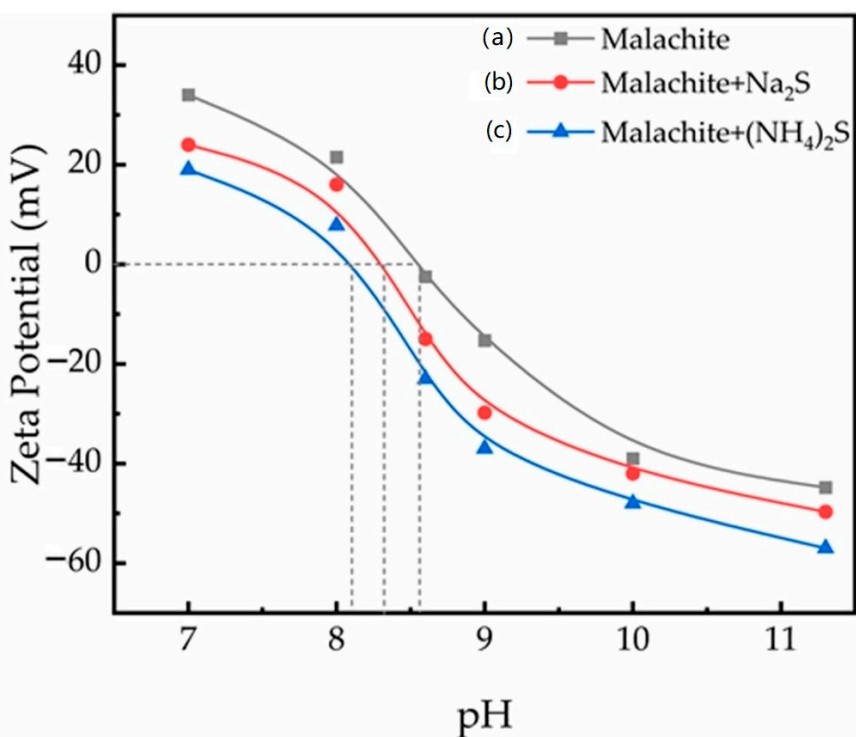

**Figure 3.** Zeta potential as follows: (**a**) malachite, (**b**) malachite + Na$_2$S and (**c**) malachite + (NH$_4$)$_2$S as a function of pH, Na$_2$S = 6 × 10$^{-2}$ mol/L, (NH$_4$)$_2$S = 6 × 10$^{-2}$ mol/L.

*3.3. FTIR Analysis*

Figures 4 and 5 show the FTIR spectra of (NH$_4$)$_2$S, Na$_2$S, and the malachite treated with Na$_2$S or (NH$_4$)$_2$S. The combined copper atoms with N and S atoms were deduced by the adsorption band of the S−H and N−H groups in ammonium sulfide during the (NH$_4$)$_2$S reaction to cupric ions in solutions, and C−O−Cu, N−O−Cu bonds. In Figure 4a, the sodium sulfide FTIR spectrum revealed one adsorption band at 3416 cm$^{-1}$, which corresponded to the typical peaks of the O−H groups, respectively. The peaks at 2993 and 2920 cm$^{-1}$ appeared after being treated with sodium sulfide and were assigned to the stretching vibration of the C−H group. The intensive bands located at 1874, 1567 and 1056 cm$^{-1}$ were correlated with the C−O, C−N and C=S groups, as shown in Figure 5b [16,17]. Figure 5a shows pure malachite, and Figure 4b displays (NH$_4$)$_2$S. The FTIR spectrum revealed two adsorption peaks at 3442 and 3199 cm$^{-1}$, which linked to the characteristic bands of S−H and N−H groups, whereas Figure 5c demonstrates characteristic malachite surface NH$_2$, C−H, C=O, C−O−C and C−S peaks at approximately 3497, 2996, 2922, 1705 and 1168 cm$^{-1}$, respectively, suggesting that (NH$_4$)$_2$S was adsorbed on the malachite surface; this indicates that the electron cloud density had moved from the C−S group to the Cu atom to form Cu−S bonds on the malachite surfaces [18].

Compared with the malachite treated with sodium sulfide, the malachite treated with ammonium sulfide had longer wavenumbers, and the wavenumbers shifted considerably. Furthermore, when comparing Figure 5b,c with the FTIR spectrum of the malachite, Figure 5a shows how new peaks might clearly be observed ((2993, 2920 and 1694 cm$^{-1}$), (2997, 2920 and 1705 cm$^{-1}$)), which appear in the FTIR spectrum after the malachite was treated with Na$_2$S or (NH$_4$)$_2$S, indicating the adsorption of sodium sulfide and ammonium sulfide on the malachite surface. Meanwhile, the FTIR spectra of malachite with sodium sulfide and with ammonium sulfide treatment were substantially different, demonstrating a strong interaction between ammonium sulfide and, therefore, more stability than the interaction between the malachite and sodium sulfide.

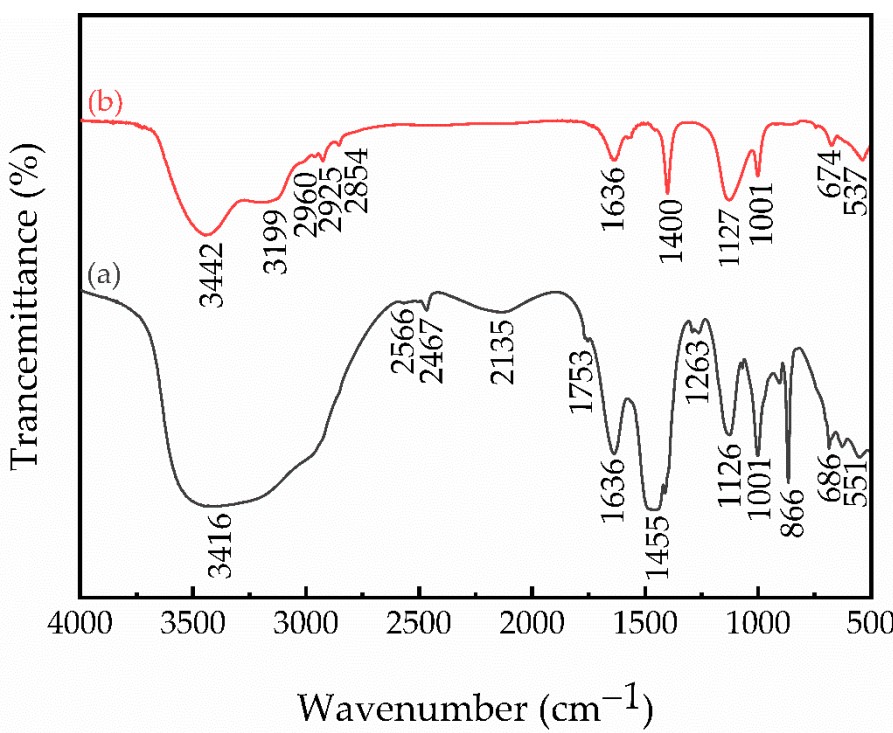

**Figure 4.** FTIR spectra of (**a**) sodium sulfide and (**b**) ammonium sulfide.

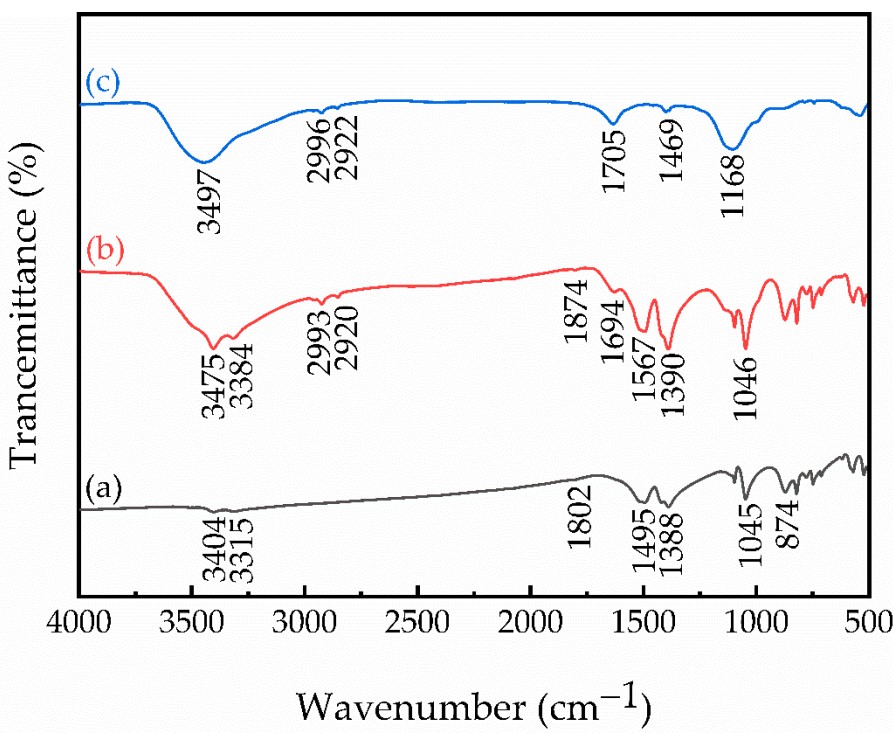

**Figure 5.** FTIR spectra as follows: (**a**) pure malachite, (**b**) malachite + $Na_2S$ and (**c**) malachite + $(NH_4)_2S$, $Na_2S = 6 \times 10^{-2}$ mol/L, $(NH_4)_2S = 6 \times 10^{-2}$ mol/L.

*3.4. XPS Analysis*

The surface elemental composition and chemical states might be effectively determined by using XPS analysis [19]. In this research, XPS measurements were also implemented to evaluate the surface of the malachite and to characterize the effect of $(NH_4)_2S$ on the surface composition and malachite sulfidization.

Figure 6a shows how the C1s spectra of malachite, when treated with Na$_2$S, consisted of three spectral peaks: C1s at ~284.80 was ascribed to C−C, C=C and C−H; C$_2$ at 286.30 eV was allocated to C−O single bonds; C$_3$ at 289.69 and 289.51eV were ascribed to O−C=O, namely, carbonate in the malachite crystal structure [20,21].

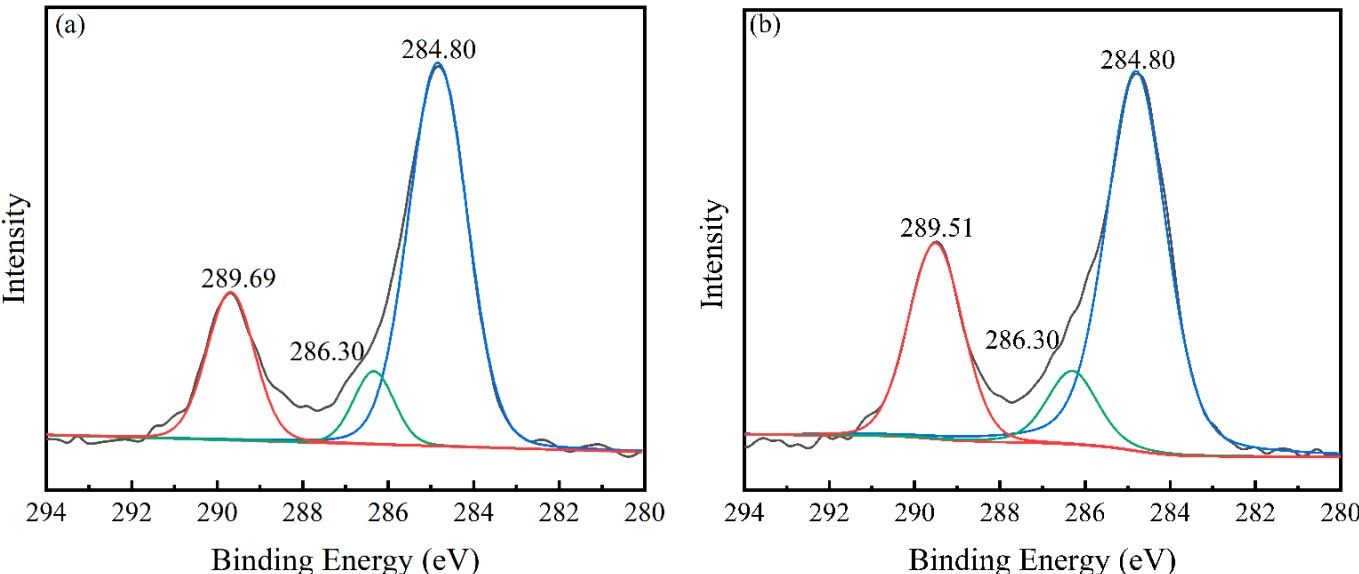

**Figure 6.** C1s XPS spectra of malachite surfaces as follows: (**a**) treated with Na$_2$S, (**b**) treated with (NH$_4$)$_2$S, Na$_2$S = 6 × 10$^{-2}$ mol/L, (NH$_4$)$_2$S = 6 × 10$^{-2}$ mol/L.

The malachite sample was treated with Na$_2$S; The peak of O1s located in 531.43 eV was attributed to the hydroxide, and then another peak located in 532.70 eV was from the carbonates.

While the ammonium sulfide was used to treat the malachite sample, S ions and solution decantation, the binding energies of the O1s spectrum peaks shifted to 531.41 eV and 532.47 eV in Figure 7b. Compared with Figure 7a, the hydroxyl species' binding energy of oxygen changed significantly (0.18 eV). To explain this, the oxygen types on the malachite surface were processed by the interaction of NH$_4$$^+$ ions with S ions. As a result of this phenomenon, the chemical environment of the O species was altered.

The S 2p spectrum was then studied further to determine the influence of (NH$_4$)$_2$S and Na$_2$S on the sulfidization products on the malachite surface, and the findings are shown in Figure 8a,b and Table 2, which display the S 2p spectra of the malachite after 6 × 10$^{-2}$ mol/L Na$_2$S treatment. The S 2p3/2 peak was provided with two spectral peaks; In addition, the binding energies were reported to be 161.67 eV and 162.85 eV, respectively. A total of 75.63% of the species in S$^{2-}$, 15.34% of the species in S$_n$$^{-2}$, and 9.03% of the species in SO$_n$$^{-2}$ were detected. These findings indicate that after sulfurizing the malachite, the S$^{2-}$ and S$_n$$^{2-}$ species can adsorb on the sample surface [22,23]. As can be seen in Figure 8b, the S 2p spectrum of the sample, which was sulfidized with 6 × 10$^{-2}$ mol/L (NH$_4$)$_2$S, was fitted with two contributions: the first one had a binding energy of 162.51 eV for the S$^{2-}$ level, the second seemed to have a binding energy of 164.49 eV for the S$_n$$^{2-}$ level, respectively. A total of 66.18% of the species in S$^{2-}$, 21.07% of the species in S$_n$$^{2-}$ and 12.75% of species in SO$_n$$^{-2}$ were accounted for and are, therefore, credited to the polysulfide ion (S$_n$$^{2-}$, n ≥ 2) [24].

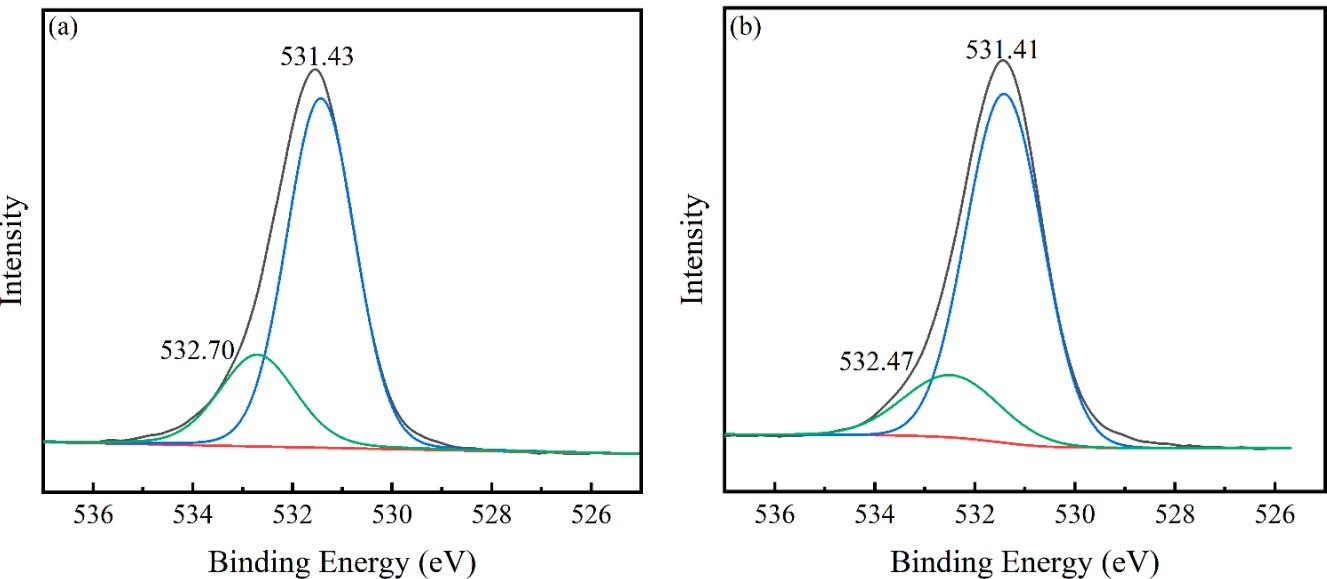

**Figure 7.** O1s XPS spectra of malachite surfaces as follows: (**a**) treated with $Na_2S$, (**b**) treated with $(NH_4)_2S$, $Na_2S = 6 \times 10^{-2}$ mol/L, $(NH_4)_2S = 6 \times 10^{-2}$ mol/L.

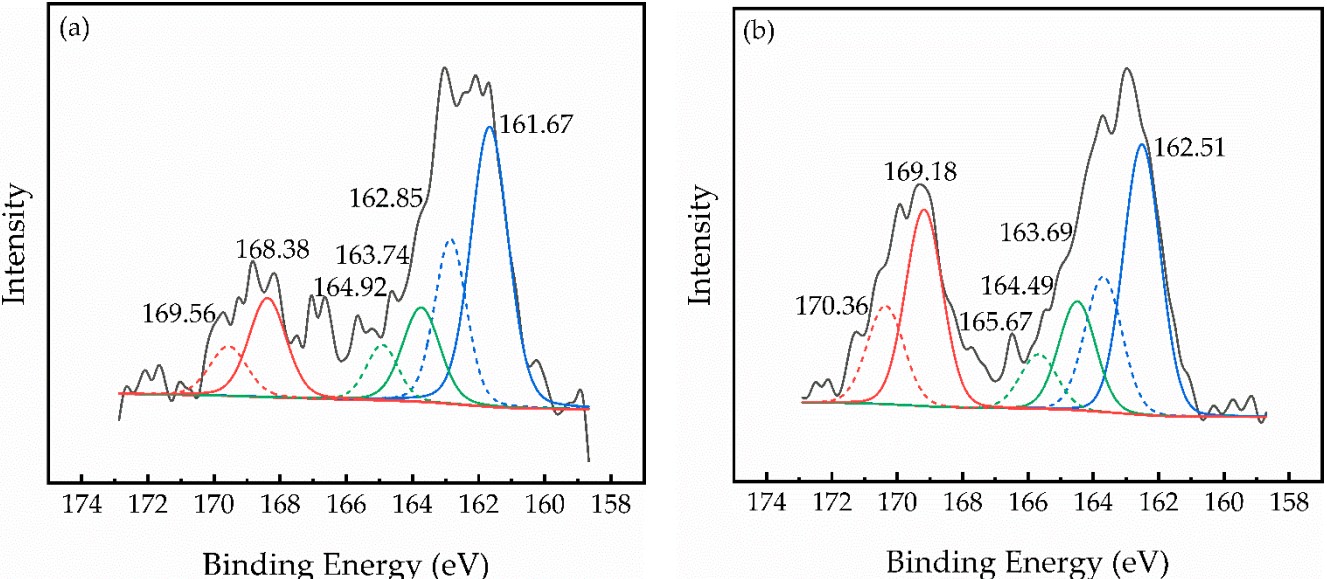

**Figure 8.** S 2p XPS spectra of malachite surfaces as follows: (**a**) treated with $Na_2S$, (**b**) treated with $(NH_4)_2S$, $Na_2S = 6 \times 10^{-2}$ mol/L, $(NH_4)_2S = 6 \times 10^{-2}$ mol/L.

The binding energies, which consist of the S 2p3/2 and S 2p1/2 doublet located at 161.67–162.51, 163.74–164.49, 164.92–165.67, and 168.38–169.18 eV, were credited to $S^{2-}$, $S_n^{2-}$ and $SO_n^{2-}$ [25,26].

Figure 9 demonstrates how two peaks of Cu 2p3/2 and Cu 2p1/2 doublet were fitted for the malachite spectra after $Na_2S$ or $(NH_4)_2S$ treatment. After pretreatment with $Na_2S$, the Cu 2p spectrum and binding energy was 932.65 eV for Cu 2p3/2 and 934.94 eV for Cu 2p1/2, which is shown in Figure 8a [22]. Figure 8b illustrates the Cu 2p spectra of malachite after $(NH_4)_2S$ was added and the binding energy at 933.03 eV for Cu 2p3/2, as well as two different pairs of spin-orbit split peaks in which Cu 2p3/2 and Cu 2p1/2 doublets were generated by MultiPak software. It is possible that the new peak of 933.03 eV may be due to Cu (I) species, while the energy band of 935.23 eV could be assumed to be due to Cu (II) species. This implies that after $Na_2S$ was added, the chemical environment of copper ions

on the malachite surface was altered, as well as the reduction of Cu (II) ionic species to Cu (I), resulting in cuprous oxide.

**Table 2.** Binding energy and percentage in total S of different S species on (a) malachite + 0.06 M $Na_2S$, (b) malachite + 0.06 M $(NH_4)_2S$.

| Sample | Specie's | S 2p Binding Energy, eV | At. % | Percentage in Total S, % |
|---|---|---|---|---|
| a | $S^{2-}$ | 161.36 | 2.87 | 75.63 |
| | $S_n^{2-}$ | 163.94 | 2.41 | 15.34 |
| | $SO_n^{2-}$ | 167.77 | 0.35 | 9.03 |
| b | $S^{2-}$ | 161.98 | 2.94 | 66.18 |
| | $S_n^{2-}$ | 164.10 | 2.58 | 21.07 |
| | $SO_n^{2-}$ | 168.69 | 0.47 | 12.75 |

When compared to Figure 9a, it might be observed that the binding energy of Cu (I) was changed and the proportion of Cu (I) species to the overall Cu was increased. This occurrence illustrates that the chemical state of Cu (I) species was altered, as well as an increase in Cu (I) species, signifying the greater extent of sulfidation on the malachite.

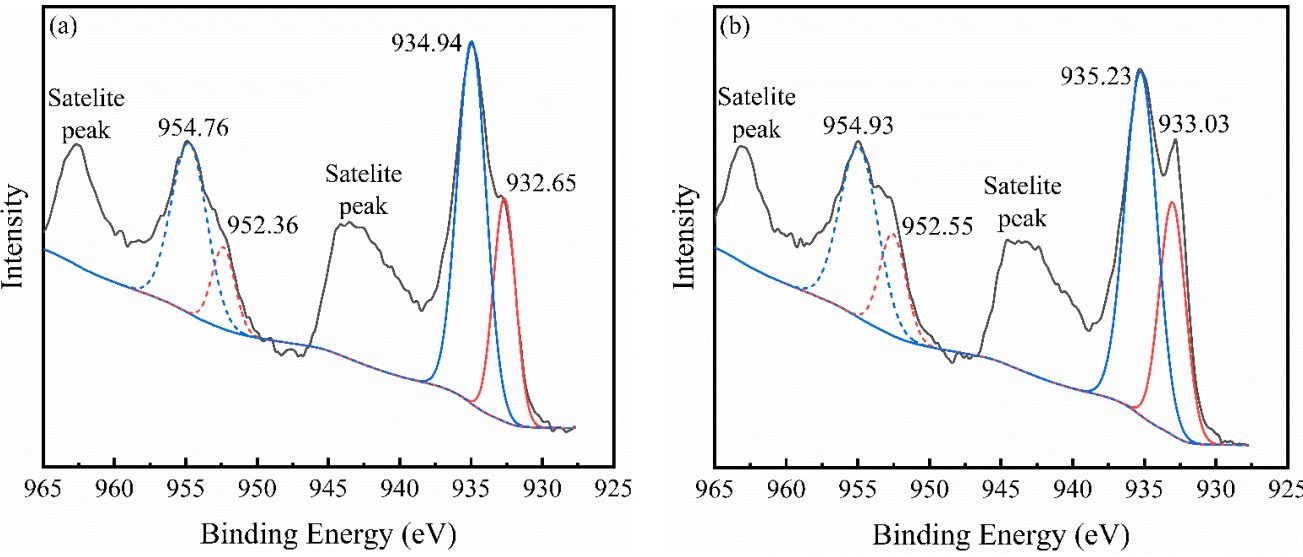

**Figure 9.** Cu 2p XPS spectra of malachite surfaces as follows: (**a**) treated with $Na_2S$, (**b**) treated with $(NH_4)_2S$, $Na_2S = 6 \times 10^{-2}$ mol/L, $(NH_4)_2S = 6 \times 10^{-2}$ mol/L.

Figure 10b shows that when the malachite was treated only with 0.06 M $Na_2S$ solution, no N peaks occurred upon the surface, and only the typical peaks of C1s, O1s, Cu3s, and Cu 3p occurred, demonstrating that $S^{2-}$ did not absorb on the malachite surface. Figure 10c shows that it was not possible to detect the N peak (derived from $NH_4^+$) on the malachite surface following treatment with 0.06 M $(NH_4)_2S$ solution due to the formation of sulfide compounds, which were absorbed on the malachite surface, and inhibit copper ions from dissolving [28]. According to XPS measurements, neither ammonium ($NH_4^+$) nor sulfide ($S^{2-}$) was adsorbed onto the malachite surface. Therefore, the data indicated that $(NH_4)_2S$ is merely a transition state, while the reactions that occurred in the solution were essential to refloat the malachite in excess $Na_2S$.

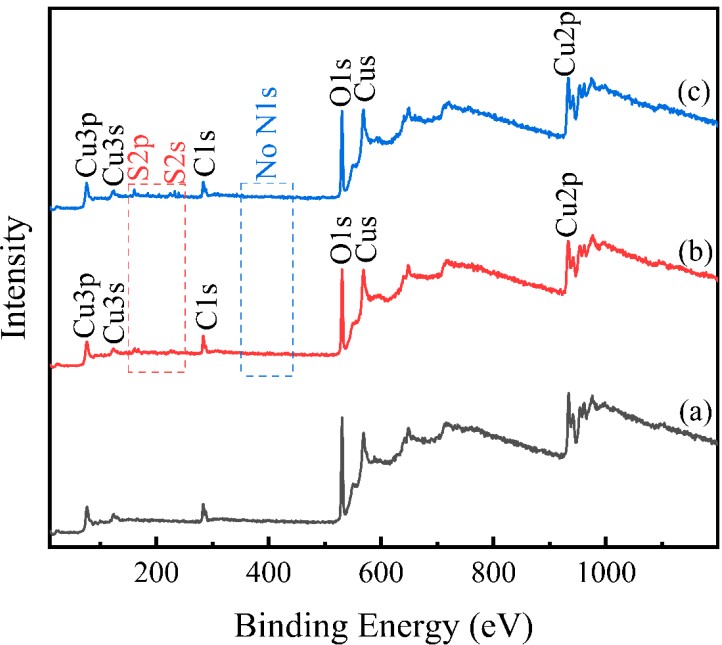

**Figure 10.** XPS survey spectra as follows: (**a**) pure malachite, (**b**) treated with $Na_2S$ and (**c**) treated with $(NH_4)_2S$, $Na_2S = 6 \times 10^{-2}$ mol/L, $(NH_4)_2S = 6 \times 10^{-2}$ mol/L.

*3.5. TOF–SIMS Analysis*

Numerous research studies have demonstrated that TOF–SIMS provides the sensitivity required for detecting and analyzing mineral surfaces from the flotation process [29,30]. ToF–SIMS was used in this study to chemically analyze the malachite surfaces. Malachite samples were treated with varying concentrations of $(NH_4)_2S$ and $Na_2S$. Figure 11a shows the malachite sample images treated with sodium sulfide. Figure 11b illustrates the images of malachite samples treated with ammonium sulfide, and negative $S^-$, $S_2^-$, $CO_3^-$, and $SO_3^-$ ions were identified on malachite surfaces that were treated with different doses of $Na_2S$ at a pH of 8.7. The distribution of S ions on the surfaces was heterogeneous, with $S^-$, $S_2^-$, and $SO_3^-$ ions that were obvious at the surface of the malachite. Ion intensity was higher in the area with a brighter color, and lower in the area with a darker color.

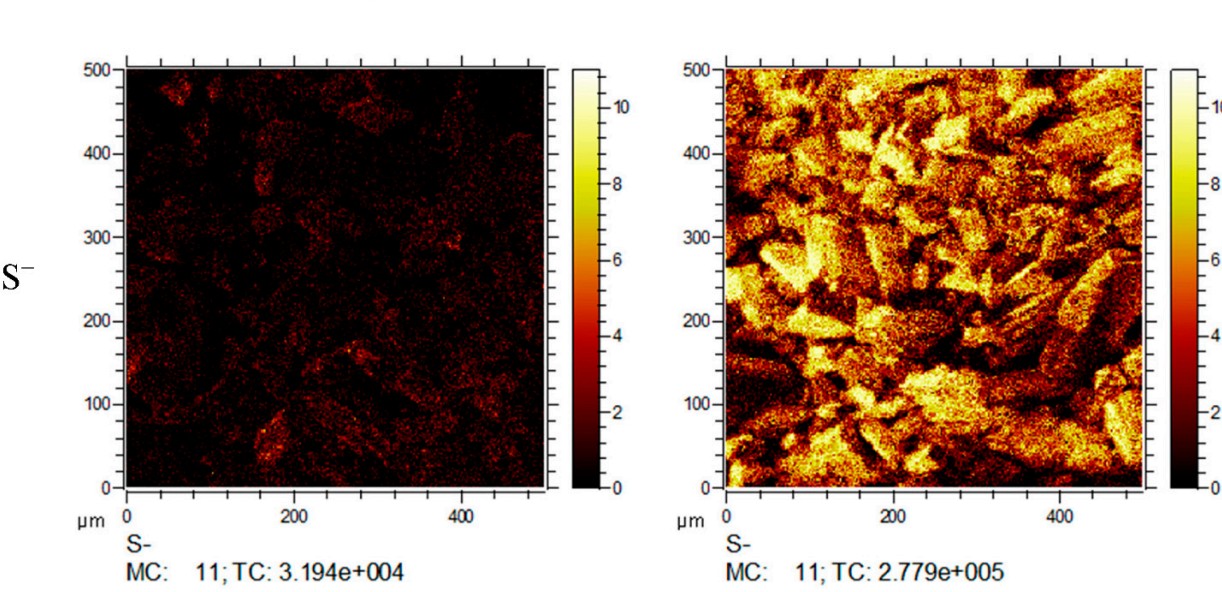

**Figure 11.** *Cont.*

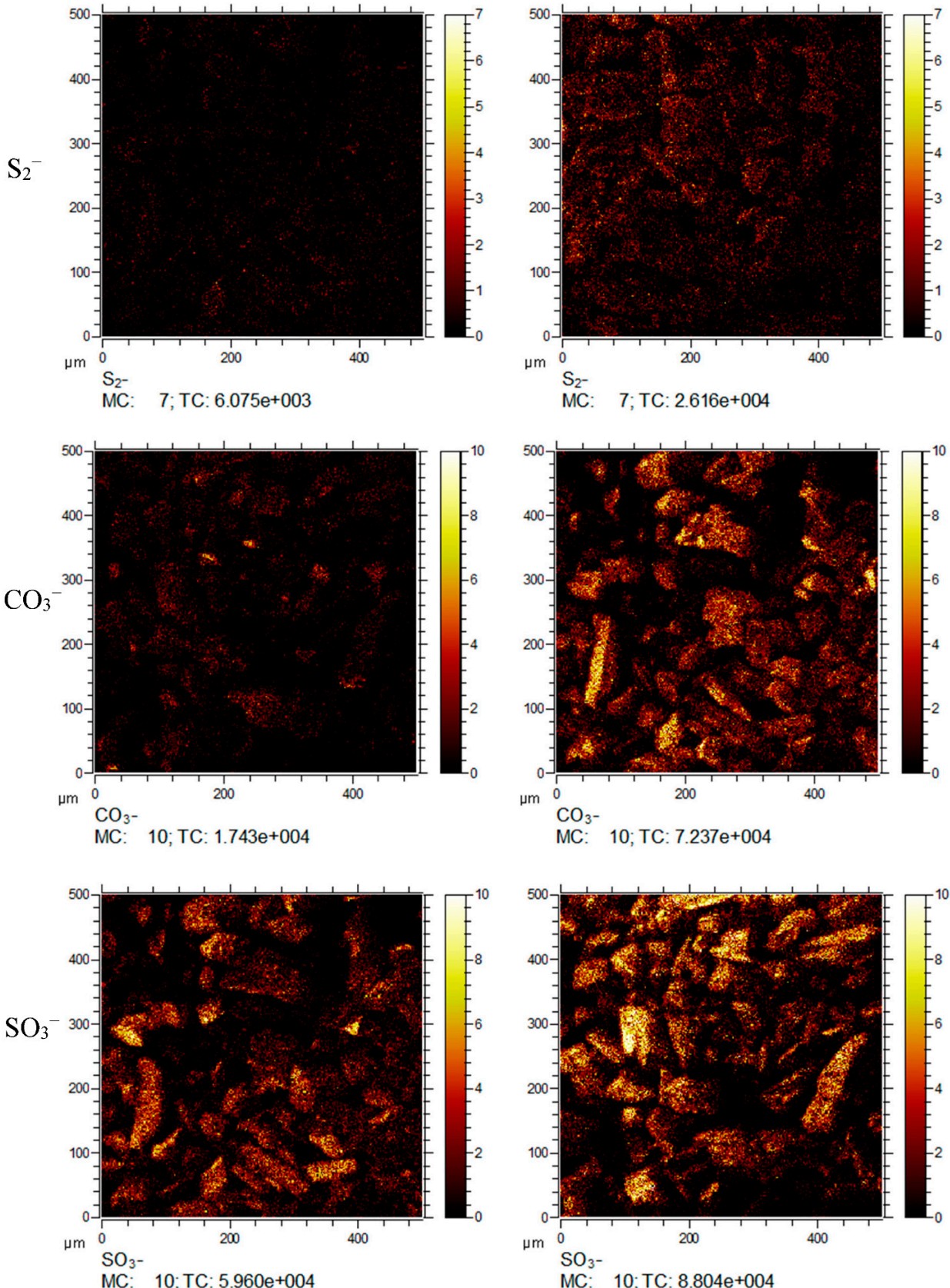

**Figure 11.** ToF–SIMS negative ion images of the malachite surface as follows: (**a**) treated with Na$_2$S, and (**b**) treated with (NH$_4$)$_2$S, Na$_2$S = 6 × 10$^{-2}$ mol/L, (NH$_4$)$_2$S = 6 × 10$^{-2}$ mol/L.

The ion intensity was significantly negatively correlated to the molecule content or element; The area with surface depression had a low ion intensity. Moreover, when the sample was treated with $(NH_4)_2S$ at the same pH, negative ion species such as $S^-$, $S_2^-$, and $SO_3^-$ became more broadly distributed over the surface of the malachite. After the addition of ammonium sulfide, the intensity of sulfur elements on the malachite surface was enhanced. In addition, remarkably, after the $Na_2S$ treatment, a small quantity of $CO_3^-$ was scattered on the malachite surface, whereas, on the malachite that was exposed to $(NH_4)_2S$ concentration, only a small amount of this ion could be found on the malachite surface. In particular, this indicates that the $CO_3^-$, which is linked to the amount of sulfurization that occurs on the malachite surface, was weakly visible. Ultimately, more sulfur species were formed on the malachite surface after it was exposed to $(NH_4)_2S$, and more copper oxide was changed into copper sulfide species and, accordingly, the malachite's floatability increased. Furthermore, according to characterization and flotation results, the increasing hydrophobicity of the malachite surface generated by $Na_2S$ and $(NH_4)_2S$ is described briefly and displayed in the graphical model in Figure 12.

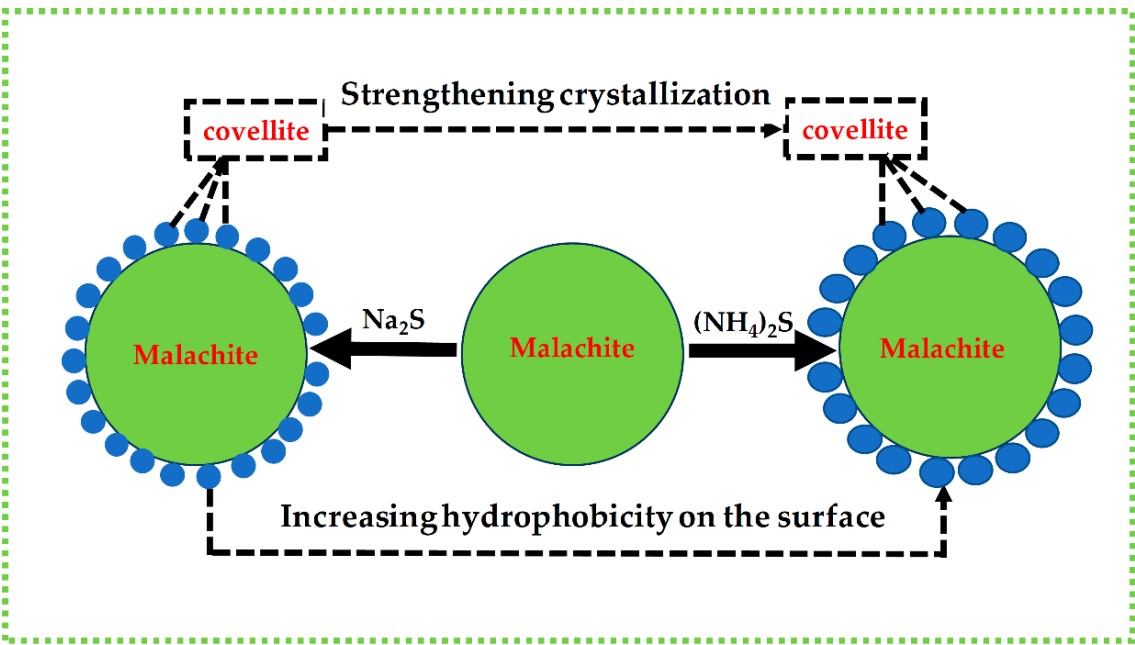

**Figure 12.** Graphical model of increasing hydrophobicity on the malachite surface by sodium sulfide and ammonium sulfide.

### 4. Conclusions

This study looked into the effect of ammonium sulfide on the sulfidization of malachite flotation, and the results are as follows:

- Micro-flotation experiments revealed that ammonium sulfide aids in the sulfidization of malachite surface flotation more than when we added sodium sulfide. Thus, the recovery of copper increased to 84%.
- The addition of ammonium sulfide led to less IEP compared to the addition of sodium sulfide, according to the research results of the Zeta tests.
- According to FTIR analyses, ammonium sulfide and sodium sulfide had an effective role in malachite sulfurization. Moreover, the FTIR analysis showed that a new characteristic peak at 1694 $cm^{-1}$ on the malachite surfaces, after being treated with sodium sulfide, indicated that Cu-S bonds had formed on the malachite surfaces.
- Malachite surfaces that had been pretreated with ammonium sulfide were activated, which led to an increase in the quantity of S species adsorbed on the malachite surface, thus enhancing its floatability. Furthermore, the XPS findings indicated that

> Cu (I)/Cu (II) mixed-surface compounds composed of $(NH_4)_2S$ were stabilized on the malachite surfaces via Cu (I)−S and Cu (II)−O bonds.
> - ToF–SIMS analysis revealed that $(NH_4)_2S$ was formed on malachite surfaces, accompanied by the formation of Cu−S.

**Author Contributions:** Conceptualization, A.M.I., D.L. and P.S.; methodology, A.M.I., J.C., P.S. and D.L.; software, A.M.I. and D.L.; validation, A.M.I., J.C., P.S. and D.L.; writing—original draft preparation, A.M.I.; writing—review and editing, D.L.; supervision, J.C., P.S. and D.L.; funding acquisition, P.S. and D.L.; Data curation and visualization, J.C., C.S. and X.J. All authors have read and agreed to the published version of the manuscript.

**Funding:** This research project was supported by the National Natural Science Foundation of China (Grant No. 52074138), Yunnan Major Scientific and Technological Projects (Grant No. 202202AG050015), Basic research project of Yunnan Province (Grant No. 202001AS070030 and 202201AU070099).

**Conflicts of Interest:** The authors declare no conflict of interest.

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
