# Peer review of "Effect of Ammonium Sulfide on Sulfidization Flotation of Malachite"

_minerals, doi:10.3390/min12101193_

Round 1
Reviewer 1 Report
This show that various test for the physico-chemical properties were used to elucidate the effect of ammonium sulfide on the flotation of malachite. The direction to find the correlation between each experiment and to increase the reliability is highly appreciated. However, it is judged that more concrete emphasis on novelty is necessary. Some critical information is missing in current form as suggested below.
1. Lines 53-54: the authors are mentioned about the defect of Sodium sulfide and ammonium sulfide such a toxicity in line 47-48. However, despite the use of these two reagents, the differences from previous studies I would very much like to ask the authors to define/emphasis the novelty of their work more clearly.
2. It would be better if you can add the error bar in Recovery data to improve the reliability.
3. Line 177, It is not clear what the comparison data with Figure 4(a) is. More detail explanation is needed.
4. It would be good to add an explanation to the caption of the meaning of each line color in Figure 7. Overall, the description in the caption for the figure is insufficient.
5. Table2 is missing.
6. In the case of Figures 8 and 9, it will be helpful to understand the difference between the solid line and the dotted line in the caption.
7. Unification of font size (In the case of caption of Figure 1), need to confirm subscript typo.
8. A unified terminology is required.
Author Response
Please see the attachment in the box

Reviewer 2 Report
This paper studied the effect mechanism of malachite surface modified by sodium sulfide and ammonium sulfide. The authors use FTIR analysis, XPS, ToF-SIMS and Zeta potential measurement to explore the interaction of (NH4)2S and Na2S with the malachite surface and follow the observations with flotation experiments varying concentration of the reagent. I think it needs a revision before it is accepted, mainly because of some issues:
1. The abstract should highlight the originality, innovation and uniqueness of this paper, instead of listing test methods and data.
2. There are many impurity peaks that are not marked in Fig.1, many impurity peaks are not marked, describe them or explain the reasons.
3. The direct flotation recovery of malachite as a variable of NaBX concentrations should be added in Fig.2 as a comparison.
4. The surface of malachite has strong solubility, which is an important reason for the low recovery of direct flotation. Whether the addition of ammonium sulfide will strengthen the stability of malachite surface?
5. Would it be possible to describe the effect mechanism with a graphical model?
Author Response
Please see the attachment in the box

Reviewer 3 Report
The authors investigated the feasibility and advantages of using ammonium sulfide instead of sodium sulfide as malachite sulfurization agent. I think the paper can be accepted after the following revision.
1. Mistakes in grammar, capitalization, font size, etc. should be carefully corrected before resubmitting the paper.
2. Table 1. should not be in the form of a picture.
3. It is recommended that sodium sulfide and ammonium sulfide be compared under optimal conditions.
4. Lines 142-143: With different dosages of NaBX concentrations, the recovery of sulfidized malachite presence of ammonium sulfide washigher than that absence of ammonium sulfide. Ammonium sulfide has a positive influence on malachite sulfidation flotation, according to these results. The expression is ambiguous and should be a comparison of sodium sulfide and ammonium sulfide, not the presence or absence of ammonium sulfide.
3. The FTIR testing procedure for sodium and ammonium sulfide in the study method is not mentioned.
4. The pH and sodium sulfide were mentioned during the ToF-SIMS, but the ph of ammonium sulfide was not mentioned. Moreover, what is the reason for choosing these ph?
5. Lines 277-279: In addition, The CO3- component progressively decreases, implying that adding ammonium-sulfide to the malachite surface could enhance sulfurization behavior. It is recommended that the author address this statement in detail. The test results do not seem to support this statement.
6. The current discussion is very bad. The authors are suggested to discuss fully with references or research results.
Author Response
Please see the attachment in the box

Round 2
Reviewer 1 Report
All my comments have been well addressed, the revised manuscript can be accepted. But I have one minor comment as follows:
It would be nice if novelty was emphasized a little more when compared with previous literature.
Author Response
Please see the attachment in the box

Reviewer 3 Report
I think the paper can be accepted.

Author Response
Please see the attachment in the box
